# Concurrent Validity and Reliability of Two Mobile Phone Applications for Measuring Vertical Jumps in Amateur Handball Players

**DOI:** 10.3390/jfmk10020223

**Published:** 2025-06-09

**Authors:** Amândio Dias, Alexandre Coutan, Bruno Silva, Catarina Eufrásio, Maria Teixeira, Mariana Alberto

**Affiliations:** 1Integrative Movement and Networking Systems Laboratory (INMOVNET-LAB), Egas Moniz Center for Interdisciplinary Research (CiiEM), Egas Moniz School of Health & Science, 2829-511 Caparica, Portugal; 2Egas Moniz Center for Interdisciplinary Research (CiiEM), Egas Moniz School of Health & Science, 2829-511 Caparica, Portugal; 3Sport Physical Activity and Health Research & Innovation Center (SPRINT), 2040-413 Rio Maior, Portugal; 4Physio-mob Clinic, 1163 Etoy, Switzerland; 5Lambert Clinic, 1750-071 Lisbon, Portugal

**Keywords:** athletes, contact platform, mobile application, jump height, validation study

## Abstract

Objectives: This study compares My Jump Lab and VertVision apps for measuring vertical jump height in handball players, assessing their validity and reliability. The research assesses both apps’ accuracy, particularly concerning higher jumps, where errors have been noted, aiming to determine the most reliable tool. The goal is to provide a direct comparison between apps, as well as confirm the apps’ validity and reliability for handball players. Methods: The present research is a transversal observational study. Twenty-seven amateur athletes performed five jumps on a contact mat with simultaneous recording by a smartphone with a high-speed camera. Pearson’s r, ICC, SEM, CV, and Bland–Altman plots were used to evaluate discrepancies and determine accuracy. Results: Pearson correlation showed strong relationships, with ICC values between 0.993 and 0.998. Both apps overestimated jump height by 1.86% compared to the platform. Bland–Altman plots indicated minimal differences between observers, confirming high validity and reliability for CMJ measurement. Both apps demonstrated very high concurrent validity (*r* > 0.9) and reliability, with ICC values near 1 and CV below 5%. My Jump Lab exhibited smaller inter-observer differences, indicating greater consistency. Conclusions: The ease of use, affordability, and portability make both apps valuable for performance monitoring, training, and injury recovery. While both demonstrated good validity and reliability, My Jump Lab proved more consistent in jump comparisons. These tools extend beyond sports, supporting physiotherapy assessments and athletic training across diverse populations.

## 1. Introduction

Measurement of countermovement jumps (CMJs) is one of the oldest and most common methods to determine physical capabilities such as measures of neuromuscular performance, fatigue, metabolic markers of exercise performance, and psychobiological indices of effort (exhaustion and stress) related to the success of athletes in competitions [1,2,3,4,5,6]. Force platforms, accelerometers, and contact mats are the most accurate methods for this type of assessment [1]. However, these instruments are regularly difficult to access for different populations, being usually used in a laboratory environment and requiring specific software for data interpretation [1,7,8,9,10].

Force platforms can measure vertical jump height using both times in the air and takeoff velocity methods. While takeoff velocity is considered the most accurate method for measuring vertical jump height, the time in the air method has been proven to be highly valid and reliable too, and most instrumental nowadays, by calculating jump height trough measurement of the jump’s flight time [9].

In recent years, due to the popularity, portability, accessibility, and advanced technology of mobile phones, several smartphone applications were created to measure variables related to physical performance and health. That’s how the My Jump Lab and VertVision apps came about [3,4,8,11,12,13]. Smartphones can provide the ideal device for testing, since they are easy to transport and can handle a high level of technology and connectivity, allowing for real time assessments.

Recent studies have shown the high reliability and accuracy of the previous version of My Jump Lab mobile application compared to the force platform in different populations, namely healthy adults [3,4], elderly people [7], and football athletes with cerebral palsy [14]. This is an affordable and portable alternative to other tools that assess vertical jump performance [6,8,11,12,15]. However, there is a lack of evidence concerning athletes who jump regularly for performance and the use of smartphone applications to assess those jumps.

Some studies have focused on the My Jump Lab application’s validity to measure an average person’s jump height [9,11,16,17]. In these groups of people, it was observed that the higher the person jumped, the greater the error in the measurement of the jump compared to the reference method used [17]. The VertVision smartphone application works in a similar manner to My Jump Lab to measure jump height. It uses frame-by-frame video analysis to determine jump start and finish and estimate jump height. However, contrary to My Jump Lab, which has multiple studies that determined its validity and reliability, the VertVision app only presents one study that focuses on its validation [18].

Nonetheless, the data available in the literature regarding the validity of these applications with athletes is scarce. The need for more studies on validity is paramount to ensure that the applications measure what they claim to measure. Additionally, reliability is also a key factor to assess, to ensure consistent results over time. If these psychometric evaluations are not met, decisions made from the provided data could be flawed—impacting athlete performance assessments, training decisions, or injury prevention strategies.

In handball, the ability to jump is important for success, and it was demonstrated that handball athletes jump higher than most of the population [19]. In this sport, it is evident that accurately assessing a handball athlete’s jumping height can be beneficial in identifying individuals who may be at greater risk of suffering related injuries, due to inadequate jumping mechanics [20]. This specific sport demands a high level of several physical elements, such as speed and agility, but jumping is the most important one, since jump throws are the most common throws [21]. Therefore, jump height can be an important factor for handball players’ performance. The need to validate these applications arises from the fact that there is no literature on this specific population, which has some specificity that could affect data from smartphone applications. Carlos-Vivas et al. [17] stated in their research that when mean jump height increases, so does the error of measurement of the smartphone app. Therefore, by using this specific group of subjects, which is characterized by higher jump height, we can ascertain if indeed there is an error of measurement and to what extent it can affect the results.

We have chosen these two specific smartphone applications because the method used for video analysis is similar and as long a common criterion is used, a direct comparison can be made between both smartphone applications. The validation will also facilitate cost, speed, and ease of evaluation of the lower limb, which, consequently, could prevent sports injuries and assist in carrying out musculoskeletal follow-up [3,4,8,11,12,13].

Thus, the present research has two main objectives: (1) to assess the two mobile applications to determine which one is more reliable and trustworthy in a training environment (usual training venue) for ecological validity by evaluating overestimation and (2) to further enhance the evidence regarding the validity and reliability of the My Jump Lab (MJ) and VertVision (VTV) mobile applications.

## 2. Materials and Methods

### 2.1. Participants and Sample Size

To ensure an adequate number of participants for our study, we reached out to two clubs via email, one club with male athletes and the other with female athletes. Both clubs are from the southern part of Lisbon. After obtaining approval from the management of both clubs, each athlete agreed to participate voluntarily in the study.

This was an observational transversal study. The number of participants was determined using a sample size calculator [21] based on the following parameters: Minimum acceptable reliability (p0) was set at 0.60, as observed in the study by Vivas [17] where the ICC did not fall below 0.60. Expected reliability (p1) was set at 0.8, assuming a higher validity index based on the previous research [9,17,22]. The significance level (α) was set at 0.05, considering a valid value if its confidence interval was above 95%. Power (1–β) was set at 80%, representing the probability of obtaining a true positive result (as commonly observed in most studies). The number of jumps per participant (K) was set at 5, deemed to strike a balance between sufficient jumps for robust evaluation and reasonable measurement time allotted to each participant [21]. The calculation revealed that a cohort of 25 participants would be necessary [21].

A total of twenty-seven subjects participated in this study, sixteen male and eleven female (age = 32.11 ± 12.9 years; weight = 77.92 ± 16.2 kg; height of 174.52 ± 10.1 cm). All athletes were amateurs and had a minimum of 5 years’ experience playing at the senior level. The inclusion criteria included female and male handball players, over 18 years old. The exclusion criteria included subjects: (a) with medical problems or a history of ankle, knee, or back pathologies in the three months prior to the study; (b) with medical or orthopedic problems that compromised their participation or performance in the study; and (c) with any reconstructive surgery of the lower limbs in the last two years or unresolved musculoskeletal disorders [4,23].

### 2.2. Instruments

A contact mat (Chronojump-Boscosystem, Barcelona, Spain,) with a system consisting of an A1-sized mat (590 × 841 mm) was used (Figure 1) as a reference method, since its validity to measure jump height has been previously established [24]. The mat is equipped with sensors and technology to accurately detect the athlete’s movements [24]. A PC-based microcontroller was connected to the mat and was responsible for calculating the flight time with a temporal resolution of 1 ms (1000 Hz), which will enable more accurate analysis of the athlete’s performance [24]. To ensure optimal sensitivity and accuracy, the platform sensitivity threshold was set at 50 ms. This threshold will define the minimum duration of the detected flight time necessary for a jump to be considered valid [4,5,7,9,10,14,21,23].

For video recording, an iPhone 15 pro max (Apple, Cupertino, CA, USA), recording at 240 frames per second, was employed. A tripod was positioned perpendicular to the frontal plane of the participants (Figure 1), centered on their feet, and placed at a distance of 1.5 m from the athletes, with a height of 30 cm [9].

### 2.3. Data Collection and Analysis

Data collection was conducted at the athletes’ training facility. Athletes were briefed on the data collection protocol and provided informed consent, with any additional questions being answered. Subsequently, lower limbs’ anthropometric measurements, specifically leg length and hip height at 90° flexion, were taken as inputs for calculation in one of the mobile phone applications and the contact mat software [1].

For each club, two different sessions were used for data collection, at the beginning of the training session. Since both clubs involve amateurs, training occurs at night, and as such, all data collection was performed in the same period of the day. Athletes were asked to arrive early at the training facility, in order to perform the tasks for our research without compromising training schedules. This would also allow for the fact that no lower-limb physical fatigue was present for the athletes at the time of data collection. All data was collected between November and December 2023, in the middle of the competitive calendar.

The warm-up routine consisted of a 10-min session of tailored low intensity running to induce a gradual increase in body temperature and prepare the cardiovascular system. Lower-limb dynamic stretching exercises were used to improve flexibility and joint range of motion. Furthermore, the participants performed three repetitions of vertical jumps to familiarize themselves with the specific type of jump mechanics under investigation [9].

All jumps were executed on a contact mat. Simultaneously they were also registered using a smartphone, as previously mentioned (Figure 1), focusing exclusively on the participants’ feet [17]. These video recordings provided detailed visual data for analysis purposes. During the CMJ, participants were instructed to place their hands on their hips, following standardized protocol. Participants completed five CMJ trials under controlled conditions, with a 1 min intertrial rest period, ensuring uniformity and minimizing potential confounding variables [9]. Athletes were instructed to jump to their maximal height. Any jump attempt that failed to meet the specified criteria (including takeoff and landing on the contact mat) was considered invalid, and participants were instructed to repeat the jump. In the background a white sheet was placed for visual contrast for data analysis. Additionally, a whiteboard with a specific code for each participant was placed, with the purpose of helping observers identify the participant and jump.

Following data collection, videos were transferred to an Android tablet Samsung galaxy A7 lite (Samsung, Seoul, South Korea) for analysis with the VTV application and an iPad Mini 5 (Apple, Cupertino, CA, USA) for analysis with the MJ application. To determine flight time with both applications, four independent observers performed the jump analysis (two for each application). A common criterion was used for all jumps: takeoff was when both feet were off the ground, and landing occurred when at least one foot touched the contact mat [9]. Both tablets facilitated independent data analysis by the observers who evaluated the jumps [3,4,9,17]. As previously mentioned, each participant was assigned a unique and specific code to ensure accurate identification and organization of the data for subsequent analysis. This coding system was consistently applied across data obtained from the contact mat and video recordings. The standardized coding allowed the observers to easily identify and compare data during the subsequent analysis phase. The reliability between observers was assessed with an intraclass correlation coefficient test, which will be further detailed in the next section.

### 2.4. Statistical Analysis

All statistical analyses were conducted using Jamovi software for statistical research (version 2.5.2) [25] with a significance level set at *p* < 0.05. To describe sample characteristics and jump performance measures, means and standard deviations were estimated. The normality of the data was assessed using the Shapiro–Wilk test to ensure statistical assumptions were met.

For assessing inter-rater reliability within each of the different calculations for the five countermovement jumps (CMJs) separately, intraclass correlation coefficients (ICCs) with a two-way random effects model was employed. Furthermore, Bland–Altman plots were employed to analyze the agreement and systematic bias, following a similar methodology described previously [26,27]. Any value of ICC < 0.5 was indicative of poor reliability, values between 0.5 and 0.75 were considered moderate reliability, values between 0.75 and 0.90 were considered good reliability, and values > 0.9 suggested excellent reliability [28]. Additionally, the coefficient of variation (CV) and standard error of measurement (SEM) were calculated to provide additional insights into the reliability of each measure. The threshold for CV was set at ≤5% [29]. In order to compare both smartphone applications, the percentage of overestimation was calculated using the equation % overestimation = 100 × (A − B)/A, where A represents the average height of the CMJ using the contact platform, and B represents the average height of the CMJ using MJ and VTV [18]. This calculation will provide insights into the level of overestimation or underestimation by the applications compared to the contact platform. With regard to the concurrent validity of MJ and VTV applications, a Pearson’s correlation coefficient (*r*) was assessed and interpreted with the following scale [30]: trivial validity < 0.10; small validity 0.10–0.30; moderate validity > 0.3–0.5; high validity > 0.50–0.70; very high validity > 0.70–0.90; and practically perfect validity > 0.90.

## 3. Results

Table 1 shows the characterization of the obtained jump data for each instrument

Table 2 shows the Pearson correlation results, which reveal high levels of concurrent validity, which range from 0.993 to 0.998. These high values indicate very strong associations between the datasets. The degrees of freedom for these associations are consistently around 132 or 133, ensuring that the estimates are both reliable and robust. This statistical assurance is further bolstered by the significant *p*-values, all of which are <0.001. Such low *p*-values strongly suggest that the results are not due to random chance but reflect true equivalence between the datasets.

Table 3 shows the ICC and confidence interval (CI) results between the contact platform (CHRJ) and all the data from the observers (MJ1, MJ2, VTV1, and VTV2), which exhibit high values signaling strong reliability across these datasets. The CI small variation further confirms these findings. For CHRJ versus MJ1, the interval ranges from 0.987 to 0.997, supporting the strong correlation. The interval between CHRJ and MJ2 is exceptionally narrow, from 0.996 to 0.998, emphasizing the stability and reliability of this correlation. Comparisons with VTV1 and VTV2 also indicate strong and reliable correlations, with intervals spanning from 0.990 to 0.996 and 0.994 to 0.998, respectively. Additionally, it also presents CV and SEM data between observers and the contact mat.

Table 4 presents the inter-rater reliability results between observers, which also present high ICC values between all of them. Additionally, the SEM is low as well as CV.

In terms of measurement precision, CVs are generally low, indicating minimal variability relative to the mean values, which speaks to the consistency of the measurements. The SEM ranges from 0.5089 cm in the MJ1 vs. MJ2 (Table 4) comparison to 1.1109 cm in the CHRJ vs. VTV1 comparison (Table 3).

Collectively, these statistical metrics not only underscore a high degree of agreement among the datasets but also demonstrate that the measurements are both valid and reliable.

My Jump Lab and VertVision generally showed an overestimation of 1.86% (Table 5) compared to Chronojump. In individual comparisons, the highest overestimation was 3.00% for the comparison between observer MJ1 and CHRJ, while the lowest was 0.68% for the comparison between observer MJ2 and CHRJ.

The applications were found to be highly valid and reliable for measuring the jump height of a CMJ relative to a contact mat.

In Figure 2 and Figure 3, several Bland–Altman plots are presented, comparing data from different observers with each other and with each instrument. The central line represents the mean absolute difference between the instruments, while the upper and lower lines represent the standard deviation and their respective confidence intervals. The data presented demonstrate that the majority of CMJ values are closely aligned with the mean of the differences between instruments or observers, indicating a high level of agreement [31]. It can be observed that both MJ and VTV slightly overestimate jump heights compared with CHRJ when jumps are less than 25 cm and slightly underestimate jump heights when jumps are greater than 25 cm, while remaining within the 95% confidence interval. Furthermore, when MJ1 and MJ2 are compared, the comparison bias is most of the time equal to 0, which indicates a strong correlation between the different measurements of the application. However, when the values are not correlated, MJ2 always overestimates the values of MJ1, which may indicate a slight positive systematic error between observer 2 and observer 1. Nevertheless, this is still small enough not to show any significant error, even if the selection of the takeoff and landing structure had to be carried out manually, which could increase the measurement error.

## 4. Discussion

The aim of this study was to compare both mobile applications to determine the more reliable, as well as provide additional evidence on the validity and reproducibility of both smartphone applications in the measurement of jump height in handball players, comparing them with a reference method.

Jumping height assessment in handball players is critical to monitoring performance, training, and injury rehabilitation [20,21]. Traditionally, the contact mat has been the standard method for this assessment. However, the platform’s cost and limited portability prevent it from being used in many contexts [20,21]. Mobile applications, such as My Jump Lab and VertVision, have emerged as promising alternatives, offering simplicity of use, low cost, and accessibility. The results of the present study are consistent with previous studies that investigated the reliability of mobile applications for measuring jump height in different populations [7,14,21,23,31,32]. The high reliability of the apps tested in our study demonstrated that they could provide consistent and accurate measurements of jump height, making them useful tools for monitoring the performance and progression of handball players.

Our study results increase previous evidence on the applications [17,18] with the iOS version of MJ and VTV having concurrent validity values, which deem a practical perfect association (*r* > 0.9) when compared to CHRJ as well as between applications. Therefore, it appears safe to assume that both smartphone applications are valid tools for measurements of the estimated jump height from the moment of flight. Both apps were found to be highly valid in measuring the jump height of the CMJ compared to a contact mat. When analyzing the inter-rater reliability of MJ and VTV applications, the results showed values close to those obtained with the contact mat. Overall, the applications had a lower minimum detectable variation in the execution of the CMJ. Results revealed a high degree of inter-rater reliability (ICC > 0.9) between observers for both apps, as well as CHRJ, with the CV value around 2/3%, which is deemed reliable, since 5% is considered the threshold level [1]. Additionally, all observers had a small SEM, lower than 1 cm. There were statistically significant differences between observers when compared to the contact mat, although higher agreement values were found between CHRJ and MJ. The overvaluation observed in all the devices remained constant, regardless of the height of the jump.

A study by Fernández et al. [9] showed, through the ICC, Pearson correlation, and CV values, the validity and reproducibility of the MJ application, which in turn is identical to our study. It has an ICC = 0.997 and a CV of less than 5%, while our study has an ICC = 0.995 and a CV of less than 5%. Pearson’s values, in the present study and in the study by Fernández et al. [9], show an almost perfect correlation between the app and the platform (*p* < 0.001). Finally, regarding the reliability of the application, almost perfect agreement was found between the difference in values between observers, with ICC values showing a difference of ±0.002 (Table 3).

A previous study [21] that shared some methodological and population similarities with ours differed in the use of instrumentation and jumping protocols. The authors used a similar contact mat as reference and compared video analysis with electromechanical sensors or accelerometers to determine jump height. Jumping was filmed with high-speed cameras, and afterward jump height was calculated using specific software, while in our study the video was used directly to determine the jump height using mobile applications. The jumping criteria were similar to those of the present study. Results indicated that the video analysis was the most accurate and reliable method for all types of jumps to determine jump height (ICC = 0.98), when compared to the contact mat [21].

It has been shown that an infrared platform has a difference of around 1.0 cm compared to a platform at 1000 Hz [27]. In addition, an accelerometric system has been shown to have an average difference of 3.6 cm compared to a platform at 1000 Hz [33], superior to both applications analyzed in this study. It seems that both applications are as reliable as these other forms of jump assessment.

The first objective of the present study was to compare both smartphone applications. An acceptable level of inter-rater reliability and concurrent validity was found for all the variables measured, but when looking at inter-rater reliability between observers (Table 4), the variance between observers (CV and SEM) of the same app was smaller on MJ than VTV. Therefore, despite both applications presenting very high inter-rater reliability and concurrent validity, MJ appears to have smaller variation for observers. In addition, both applications and the contact mat showed similar reproducibility, which is an important result considering the ecological validity of our methods.

These results have practical applications for physical therapists or strength and conditioning professionals looking for a simple approach for monitoring and measuring relevant performance variables with low-cost mobile applications. Our results revealed that VTV and MJ provide valid and reliable estimates of CMJ jump height. Despite the advantages, it is worth noting that VTV is a free app available for Android, and MJ is a paid app, present in Android and IOS operating systems. The results cannot be transferred to the game situation, and, in the current environment, this is a static jump rather than a dynamic one, which is the more common in a gaming situation.

It is important to note that, despite the high overall concurrent validity, both apps overestimated by 1.86% when compared to the contact mat, regardless of the actual height of the jump. This estimate differs from the one reported by Carlos-Vivas et al. [17], who obtained an overestimation percentage of 0.78%. The difference in overestimation may be related to the difference in methods. In our study, a tripod was used to maintain the smartphone in a stable position, while in the study of Carlos-Vivas et al. [17] the smartphone was held in the hand, which can cause a visualization error while analyzing the videos (e.g., leaning of the smartphone). Additionally, video analysis was only performed by one person, while in our study it was conducted by two different observers. So, it is possible that the increase in number of evaluators elevated the variability of video analysis and therefore increased the overestimation percentage.

The results of the study also show that the MJ and VTV apps are highly reliable for measuring the height of the CMJ jump in handball players. This was evidenced by the ICC values close to 1 for the applications and the platform, the CV of less than 5% for all measurements, the SEM of 1 cm for the applications and the platform, and the comparison using Bland–Altman plots, which showed a high degree of agreement between the different applications and observers. Compared with a similar study [17], no significant systematic bias was observed between the results of the contact platform, the applications, and the observers. Furthermore, the comparison between observers also indicated high reliability for the apps, with ICC values close to 1, CV% of less than 5%, and SEM of less than 1 cm.

These factors contribute to the practical applicability of the MJ and VTV apps in the context of handball. The apps can be used to monitor players’ performance during training and competition and to assess the progression of injury rehabilitation.

Both applications could be useful in assessing physical performance in different aspects relevant to strength and conditioning as well as physiotherapy, such as strength, power, and motor control. The fact that apps can be used as a practical measure in field assessments suggests their usefulness in non-clinical settings, where it is important to obtain quick and reliable data on patients’ physical performance.

Sport practitioners and physiotherapists can use both apps to assess recovery from sports injuries by monitoring jump height, flight time, and contact time. For physiotherapists this can help to determine the effectiveness of treatment and the gradual return to sport. During rehabilitation after orthopedic surgeries, such as anterior cruciate ligament reconstruction, the app can be used to monitor the patient’s progress in terms of muscle strength and power, helping to adapt the rehabilitation program as necessary. In cases of muscle injuries, such as sprains or strains, both apps can be used to monitor the patient’s jumping ability and detect muscle asymmetries or deficiencies during the recovery process. In addition to aiding injury recovery, the apps can be used to optimize athletic performance by providing feedback on the effectiveness of strength training and fitness programs.

Mobile phone applications, in addition to being an important tool for injury rehabilitation, are also important for injury prevention. When a unilateral assessment of the CMJ is carried out, it is possible to observe the muscular imbalances that exist between the limbs and begin to make an individualized exercise prescription for the athlete [34].

However, it is important to acknowledge the limitations of this study, such as the overestimation observed in both apps compared to the reference method, as well as the static nature of the jump assessment. Future research should aim to address these limitations by refining app algorithms to minimize overestimation and exploring the applicability of dynamic jump assessments in game-like scenarios. Additionally, more research should focus on other populations, such as the elderly population, where both apps can be used to assess physical function, which is important for preventing falls and maintaining functional independence [7,35].

Despite these limitations, the findings support the practical utility and reliability of MJ and VTV as valuable tools for jump height assessment in handball and potentially other sports and clinical settings.

## 5. Conclusions

In the present study we compared two mobile applications, namely My Jump Lab and VertVision, to ascertain which was more reliable in measuring jump height in handball players. The findings demonstrate that both applications exhibit high reliability and validity, making them valuable tools for monitoring performance and injury rehabilitation in handball players. MJ appears to be a more reliable tool, since the variance between observers was smaller. Despite slight overestimations compared to the reference method, the apps provide consistent and accurate measurements.

The study’s results underscore the practical applicability of mobile applications in sports science and physiotherapy settings. My Jump Lab and VertVision offer simplicity, accessibility, and cost-effectiveness, enabling easy monitoring of players’ performance during training and competition, as well as assessing injury progression and rehabilitation. Additionally, these apps hold promise for broader applications beyond handball, including assessing physical performance in physiotherapy, optimizing athletic training programs, and monitoring physical function in various populations, such as the elderly and individuals with chronic conditions.

## Figures and Tables

**Figure 1 jfmk-10-00223-f001:**
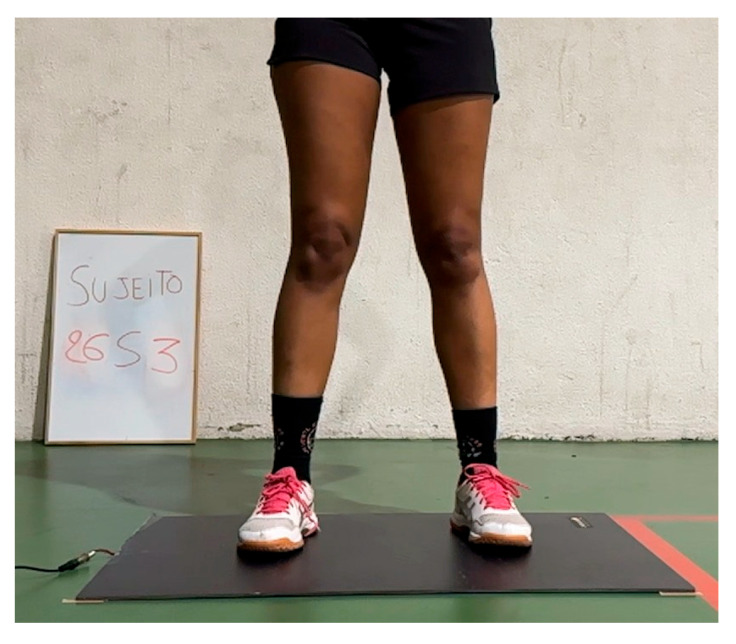
Data collection (image taken from the videos used for assessment).

**Figure 2 jfmk-10-00223-f002:**
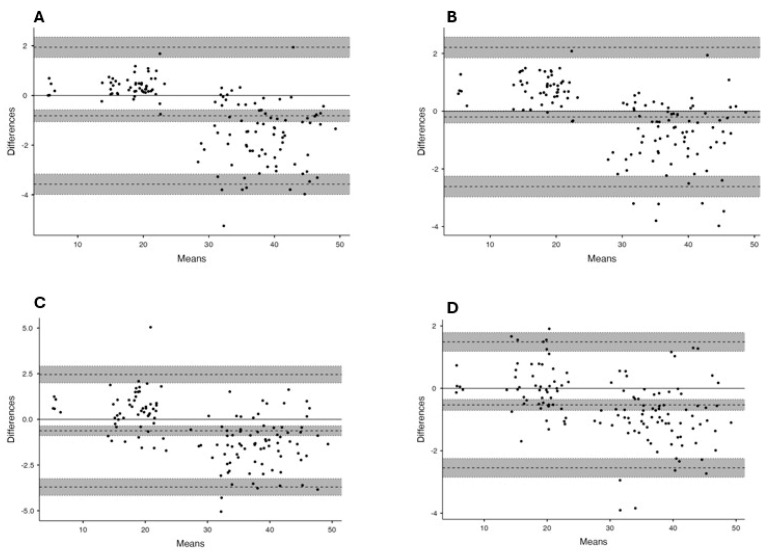
Bland–Altman plots between contact mat (CHRJ) and the 4 observers. (**A**) CHRJ-MJ1; (**B**) CHRJ-MJ2; (**C**) CHRJ-VTV1; (**D**) CHRJ-VTV2.

**Figure 3 jfmk-10-00223-f003:**
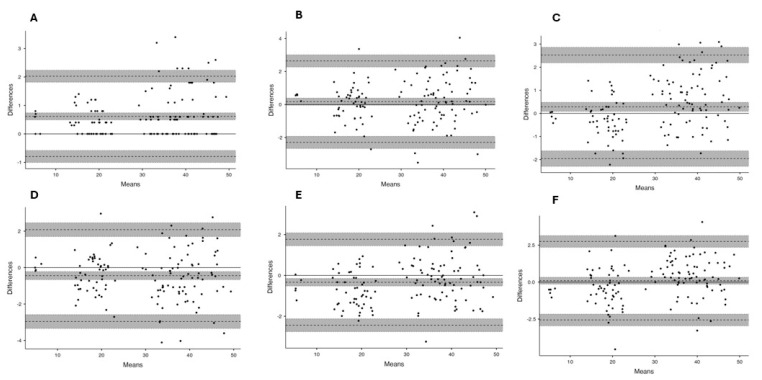
Bland–Altman plots between observers. (**A**) MJ1-MJ2; (**B**) MJ1-VTV1; (**C**) MJ1-VTV2; (**D**) MJ2-VTV2; (**E**) MJ2-VTV2; (**F**) VTV1-VTV2.

**Table 1 jfmk-10-00223-t001:** Jump performance for each instrument.

	CHRJ	MJ1	MJ2	VTV1	VTV2
mean ± standard deviation	32.1 ± 10.8	33.2 ± 11.7	32.5 ± 11.6	33.9 ± 11.6	32.8 ± 11.3

CRHJ: data from contact platform; MJ1: observer 1 for My Jump Lab; MJ2: observer 2 for My Jump Lab; VTV1: observer 1 for VertVision; VTV2: observer 2 for VertVision.

**Table 2 jfmk-10-00223-t002:** Pearson correlation results between all datasets.

		CHRJ	MJ1	MJ2	VTV1
MJ1	*r*	0.995 *			
	df	132			
MJ2	*r*	0.996 *	0.998 *		
	df	132	132		
VTV1	*r*	0.993 *	0.994 *	0.994 *	
	df	133	132	132	
VTV2	*r*	0.996 *	0.996 *	0.995 *	0.993 *
	df	133	132	132	133

* *p* < 0.001; df: degrees of freedom; *r*: Pearson correlation; CRHJ: data from contact platform; MJ1: observer 1 for My Jump Lab; MJ2: observer 2 for My Jump Lab; VTV1: observer 1 for VertVision; VTV2: observer 2 for VertVision.

**Table 3 jfmk-10-00223-t003:** Inter-rater reliability between observers and contact mat.

	CHRJ vs. MJ1	CHRJ vs. MJ2	CHRJ vs. VTV1	CHRJ vs. VTV2
ICC	0.995	0.997	0.994	0.997
CI (95% lower–95% upper)	0.987–0.997	0.996–0.998	0.990–0.996	0.994–0.998
ICC interpretation	Excellent	Excellent	Excellent	Excellent
CV (%)	3.330	2.940	3.730	2.450
SEM (cm)	0.995	0.869	1.110	0.728

CRHJ: data from contact platform; MJ1: observer 1 for My Jump Lab; MJ2: observer 2 for My Jump Lab; VTV1: observer 1 for VertVision; VTV2: observer 2 for VertVision; CV%: coefficient of variation; SEM: standard error of measurement.

**Table 4 jfmk-10-00223-t004:** Inter-rater reliability between observers.

	MJ1 vs. MJ2	MJ1 vs. VTV1	MJ1 vs. VTV2	MJ2 vs. VTV1	MJ2 vs. VTV2	VTV1 vs. VTV2
ICC	0.998	0.997	0.997	0.99	0.998	0.997
CI (95% lower–95% upper)	0.991–0.999	0.996–0.998	0.996–0.998	0.99–0.996	0.996–0.998	0.995–0.998
ICC interpretation	Excellent	Excellent	Excellent	Excellent	Excellent	Excellent
CV (%)	1.569	2.940	2.680	3.040	2.540	3.180
SEM (CM)	0.508	0.889	0.808	0.910	0.758	0.956

MJ1: observer 1 for My Jump Lab; MJ2: observer 2 for My Jump Lab; VTV1: observer 1 for VertVision; VTV2: observer 2 for VertVision; CV%: coefficient of variation; SEM: standard error of measurement.

**Table 5 jfmk-10-00223-t005:** Percentage of overestimation.

	MJ1 vs. CHRJ	MJ2 vs. CHRJ	VTV1 vs. CHRJ	VTV2 vs. CHRJ	MJ TOT vs. CHRJ	VTV TOT vs. CHRJ
% Overestimation	3.00	0.68	2.03	1.69	1.86	1.86

MJ1: observer 1 for My Jump Lab; MJ2: observer 2 for My Jump Lab; VTV1: observer 1 for VertVision; VTV2: observer 2 for VertVision; MJ TOT: total number of jumps from My Jump Lab; VTV TOT: total number of jumps from VertVision.

## Data Availability

Data will be available upon request to the corresponding author.

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
