# Peer review of "Concurrent Validity and Reliability of Two Mobile Phone Applications for Measuring Vertical Jumps in Amateur Handball Players"

_jfmk, 2025, doi:10.3390/jfmk10020223_

Round 1
Reviewer 1 Report
Comments and Suggestions for Authors
1.Abstract:
There seems to be a grammatical error in the Background/Objectives section — “as well confirm” (for example, it should be “as well as confirm”).
- 2. Introduction
The second paragraph of the Introduction and the reference to Carlos-Vivas et al[17]. mention "vertical velocity at take-off and time in the air." In the article, the phrase "which are characterized by higher jump height, we can assess if indeed the error of measurement is greater when the jump is higher, as stated by Carlos-Vivas et al." should clarify what specific error of measurement is being referred to.
In the Purpose 「1) to assess/compare the two mo- bile applications, to determine which one is more reliable and trustworthy in a training environment 」“training environment” needs to be defined in terms of specific conditions (For example :competitor level).
- 3. Results:
Please check results, missing Table2
Please check the results, lack of textual explanations corresponding to Table 3.
The ICC for Table 4 is very high (0.994-0.997), but it would be better to write the ICC interpretation criteria below the graph or elsewhere (e.g., ICC > 0.9 is ‘excellent’, 0.75-0.9 is “good”). good" for 0.75-0.9.).
- 4. Discussion:
The discussion states “ Finally, regarding the reliability of the application, almost perfect agreement was found between the difference in values between observers, with ICC values showing a difference of ± 0.002 (Table 3)” is missing from the results of the article.
- Conclusions
The conclusion should succinctly summarize the main findings and their significance. For example("both apps show reliability and validity" and "useful for performance monitoring and rehab") are repeated. Combining these into single sentences would reduce redundancy and improve readability.
- 6. References
Please confirm whether the references format is correct and whether a period (.) is missing after the page number.
Author Response
Comments 1: .Abstract:There seems to be a grammatical error in the Background/Objectives section — “as well confirm” (for example, it should be “as well as confirm”).
Response 1: Thank you for your comment. We have corrected the grammatical error in the abstract.
Comments 2: 2. Introduction
The second paragraph of the Introduction and the reference to Carlos-Vivas et al[17]. mention "vertical velocity at take-off and time in the air." In the article, the phrase "which are characterized by higher jump height, we can assess if indeed the error of measurement is greater when the jump is higher, as stated by Carlos-Vivas et al." should clarify what specific error of measurement is being referred to.
Response 2: Thank you for your comment. We have added the clarification of the specific error in the text.
Comments 3: In the Purpose 「1) to assess/compare the two mobile applications, to determine which one is more reliable and trustworthy in a training environment “training environment” needs to be defined in terms of specific conditions (For example :competitor level).
Response 3: Thank you for your comment. We have specified what we meant as “training environment”.
Comments 4: 3. Results:
Please check results, missing Table 2. Please check the results, lack of textual explanations corresponding to Table 3.
Response 4: Thank you for your comment. We have added table 2 and the textual explanations for table 3. We don’t understand how that happened and are sorry for the oversight.
Comments 5: The ICC for Table 4 is very high (0.994-0.997), but it would be better to write the ICC interpretation criteria below the graph or elsewhere (e.g., ICC > 0.9 is ‘excellent’, 0.75-0.9 is “good”). good" for 0.75-0.9.).
Response 5: Thank you for your comment. We added in both table 4 and 5 a new line of information, specifically regarding ICC interpretation.
Comments 6: 4. Discussion: The discussion states “ Finally, regarding the reliability of the application, almost perfect agreement was found between the difference in values between observers, with ICC values showing a difference of ± 0.002 (Table 3)” is missing from the results of the article.
Response 6: Thank you for your comment. The information was added with the textual explanations for Table 3.
Comments 7: Conclusions
The conclusion should succinctly summarize the main findings and their significance. For example("both apps show reliability and validity" and "useful for performance monitoring and rehab") are repeated. Combining these into single sentences would reduce redundancy and improve readability.
Comments 7: Thank you for your comment. We have combined both sentences into one, as suggested.
Comments 8: 6. References
Please confirm whether the references format is correct and whether a period (.) is missing after the page number.
Response 8: Thank you for your comment. You are correct, there were some references with a period missing after the page number. We have checked all references and added the period when it was missing.
Reviewer 2 Report
Comments and Suggestions for Authors
The authors report a concurrent validity and reliability study of two mobile phone applications to measure vertical jump in handball players
They report high reliability and validity of both apps, with the My Jump app being a more reliable tool. Overall, the study is well executed, however, I am not sure about the novelty of the study goal. Concurrent validity and reliability of several mobile applications have been extensively documented. There are also some areas, such as methods and results, that can be substantially improved. I provide my recommendations below and encourage the authors to address them, which could help make the manuscript better in conveying the findings and their implications. Please find my recommendations below:
Title
Please reformulate the title by eliminating the word “comparison”. It is redundant.
Abstract
Authors should improve the abstract with the reviewer’s recommendations in the different sections of the manuscript.
Introduction
Line 39: Please change “platforms” to “mats”
Line 39-40: add a reference.
The manuscript would benefit from improved fluency and coherence in the writing, particularly in the introduction and rationale sections. Additionally, the research gap is not clearly articulated. The authors should specify why these two mobile applications were selected for analysis over other available tools, and what makes them particularly relevant for use in handball instead of other apps. Furthermore, the rationale for focusing on handball players needs to be better justified by explaining the specific performance demands or monitoring needs of this population.
Critically, the manuscript lacks a thorough analysis of previous literature. A more robust discussion of existing studies and their limitations would help contextualize the need for this research. Moreover, the importance of evaluating the psychometric properties (e.g., validity, reliability) of mobile applications in the field of sport science should be emphasized more clearly, as it underpins their credibility and practical utility for performance assessment and monitoring.
Aims: authors need to clarify the specific psychometric properties that they want to evaluate, because it is confusing to the readers. Please use only one verb to redact each objective. Do not use the slash
Materials and Methods
The authors explained in a good way the sample size calculation, and I appreciate them for doing so. However, several should recommendations should be addressed:
- Please include the study design
- Please change units of height (meters to centimeters)
Inclusion criteria:
- Please specify:
- The level of the athletes
- Years of experience
- ¿Are there athletes from handball clubs? Please mention it
- Mention the provenance of the clubs
Exclusion criterion:
- Please revise all the exclusion criteria if all of them are applicable for the participants (i.e., who were diagnosed with some type of cardiac, respiratory, or neurological pathology ¿ Could they play handball despite being diagnosed?)
- Please eliminate the (e) exclusion criteria. If someone did not sign the informed consent, they could not be excluded from the research because he/she was not included in the study.
- The procedures must be improved.
- Please consider providing greater detail regarding the timing and conditions under which the measurements were taken. Specifically, it would be helpful to clarify at what point during the season the study was conducted
- How much time was allowed between trials to assess reproducibility. ¿Three, four, or seven days?. Additionally, the type of reproducibility evaluated should be explicitly stated. Was it intra-rater, inter-rater, or test–retest reliability? This information is essential to assess the robustness and replicability of the measurement protocol.
- Authors should change Figure 1. This figure must be a good representative image of the setup of the instruments. The quality must be improved.
- The flow of the data collection and analysis section should be boosted.
Statistical analysis
- Please enhance the Jamovi software reference in the text.
RESULTS
- Please ensure that all punctuation marks (e.g., dashes, parentheses, slashes, and other symbols) are followed and/or preceded by appropriate spacing according to academic writing conventions. For example, there should be a space before and after dashes (–) when used between values or concepts. For example, p<0.05 to p < 0.05.
- Please revise the formatting and overall layout of all tables. Currently, the tables lack visual clarity and consistency, which makes it difficult to interpret the data effectively. We recommend improving the table design by ensuring proper alignment, clear column headings, consistent units, and adequate spacing. A more polished and readable table format will greatly enhance the clarity and professional appearance of the manuscript. Please be consistent using a dot or a point to separate decimals (i.e, table 4).
- Be consistent with the number of decimals.
- The quality of all the figures must be strengthened
Discussion
The manuscript appears to conflate the terms validity and reproducibility in several sections of the results and discussion. These are distinct psychometric properties, and it is important to ensure that they are used accurately and consistently throughout the text. We recommend carefully reviewing all mentions of these terms to avoid conceptual confusion and to maintain methodological clarity.
Author Response
Comments 1: The authors report a concurrent validity and reliability study of two mobile phone applications to measure vertical jump in handball players
They report high reliability and validity of both apps, with the My Jump app being a more reliable tool. Overall, the study is well executed, however, I am not sure about the novelty of the study goal. Concurrent validity and reliability of several mobile applications have been extensively documented. There are also some areas, such as methods and results, that can be substantially improved. I provide my recommendations below and encourage the authors to address them, which could help make the manuscript better in conveying the findings and their implications.
Response 1: Thank you for your comments and considerations regarding our manuscript. You are correct that several mobile applications have been documented regarding validity and reliability, but no direct comparison has been made between them, and as such, that is the primary goal of our research
Comments 2: Title. Please reformulate the title by eliminating the word “comparison”. It is redundant.
Response 2: Thank you for your comment. We have reformulated the title, as suggested.
Comments 3: Abstract. Authors should improve the abstract with the reviewer’s recommendations in the different sections of the manuscript.
Response 3: Thank you for your comment. We have updated the abstract in accordance with the recommendations for the different sections of the manuscript..
Comments 4: Introduction. Line 39: Please change “platforms” to “mats”: Line 39-40: add a reference.
Response 4: Thank you for your comment. We have changed the term used, as well added a reference in the ending of the sentence.
Comments 5: The manuscript would benefit from improved fluency and coherence in the writing, particularly in the introduction and rationale sections. Additionally, the research gap is not clearly articulated. The authors should specify why these two mobile applications were selected for analysis over other available tools, and what makes them particularly relevant for use in handball instead of other apps. Furthermore, the rationale for focusing on handball players needs to be better justified by explaining the specific performance demands or monitoring needs of this population.
Critically, the manuscript lacks a thorough analysis of previous literature. A more robust discussion of existing studies and their limitations would help contextualize the need for this research. Moreover, importance of evaluating the psychometric properties (e.g., validity, reliability) of mobile applications in the field of sport science should be emphasized
Response 5: Thank you for your comment. We added the explication on why both smartphone applications were used (similar mode for data analysis).
With regards to the specificity of the sports, we have included the rationale for utilizing handball players, since jump throws are commonly used and, as such, jump height can be a determinant factor of their performance. Additionally, previous studies have demonstrated that smartphone applications can increase their error of measurement with greater jump heights. By choosing to use handball players, which are known for jumping higher than other type of populations, we can ascertain if the error of measurement really exists and to what extent can affect results.
As to the psychometric properties, we have added a paragraph stating their importance in evaluating this type of equipment’s (mobile applications) and how they can impact athletes’ performance and even enhance injury risk.
Comments 6: Aims: authors need to clarify the specific psychometric properties that they want to evaluate, because it is confusing to the readers. Please use only one verb to redact each objective. Do not use the slash
Response 6: Thank you for your comment. We have clarified the psychometric properties that we wanted to ascertain.
Comments 7: Materials and Methods
The authors explained in a good way the sample size calculation, and I appreciate them for doing so. However, several recommendations should be addressed:
Please include the study design
Please change units of height (meters to centimeters)
Response 7: Thank you for your comment. We have added the study design (observational transversal) and corrected the units of height.
Comments 8: Inclusion criteria:
Please specify:
The level of the athletes
Years of experience?
Are there athletes from handball clubs? Please mention it
Mention the provenance of the clubs
Response 8: Thank you for your comment. We have included the level of the athletes, years of experience at the senior level and place of origin of this clubs. We did not include the name of the clubs for anonymity and in accordance with data protection laws of our country.
Comments 9: Exclusion criterion:
Please revise all the exclusion criteria if all of them are applicable for the participants (i.e., who were diagnosed with some type of cardiac, respiratory, or neurological pathology ¿ Could they play handball despite being diagnosed?)
Please eliminate the (e) exclusion criteria. If someone did not sign the informed consent, they could not be excluded from the research because he/she was not included in the study.
Response 9: Thank you for your comment. We have revised the exclusion criteria according to your suggestions.
Comments 10: The procedures must be improved.
Please consider providing greater detail regarding the timing and conditions under which the measurements were taken. Specifically, it would be helpful to clarify at what point during the season the study was conducted
Response 10: Thank you for your comment. We included a paragraph in the Data collection and analysis section addressing the conditions and timing of the measurements.
Comments 11: How much time was allowed between trials to assess reproducibility. ¿Three, four, or seven days?. Additionally, the type of reproducibility evaluated should be explicitly stated. Was it intra-rater, inter-rater, or test–retest reliability? This information is essential to assess the robustness and replicability of the measurement protocol.
Response 11: Thank you for your comment. All data was collected in two separate days, with athletes attending a single session, so no reproducibility was assessed. With regards to reliability, we included in the results section (both text and table headings) the type of reliability test (inter-rater).
Comments 12:
Authors should change Figure 1. This figure must be a good representative image of the setup of the instruments. The quality must be improved.
Response 12: Thank you for your comment. We changed the image on figure 1, to better reflect the data collection procedure.
Comments 13: The flow of the data collection and analysis section should be boosted.
Response 13: Thank you for your comment. We believe that with the information that was added to the specific section, combined with the procedures that are thoroughly explained and detailed, the flow of information for this section is adequate and equal with other studies on this topic.
Comments 14: Statistical analysis. Please enhance the Jamovi software reference in the text.
Response 14: Thank you for your comment. We have changed the reference
Comments 15: RESULTS. Please ensure that all punctuation marks (e.g., dashes, parentheses, slashes, and other symbols) are followed and/or preceded by appropriate spacing according to academic writing conventions. For example, there should be a space before and after dashes (–) when used between values or concepts. For example, p<0.05 to p < 0.05.
Response 15: Thank you for your comment. We have reviewed the results section accordingly, both in text and tables.
Comments 16: Please revise the formatting and overall layout of all tables. Currently, the tables lack visual clarity and consistency, which makes it difficult to interpret the data effectively. We recommend improving the table design by ensuring proper alignment, clear column headings, consistent units, and adequate spacing. A more polished and readable table format will greatly enhance the clarity and professional appearance of the manuscript. Please be consistent using a dot or a point to separate decimals (i.e, table 4).
Be consistent with the number of decimals.
The quality of all the figures must be strengthened
Response 16: Thank you for your comment. We have revised all tables, changed alignments and adequate spacing, as suggested, as well and consistently kept the number of decimal point equal.
New images, with greater contrast in colours were included for figures 2 and 3, as suggested.
Comments 17: Discussion
The manuscript appears to conflate the terms validity and reproducibility in several sections of the results and discussion. These are distinct psychometric properties, and it is important to ensure that they are used accurately and consistently throughout the text. We recommend carefully reviewing all mentions of these terms to avoid conceptual confusion and to maintain methodological clarity.
Response 17: Thank you for your comment. We have revised the results and discussion section and clarified the psychometric properties throughout both, in order to provide additional clarity and increase accuracy of our manuscript.
Round 2
Reviewer 1 Report
Comments and Suggestions for Authors
The revised manuscript has been appropriately revised according to the reviewers' comments. I judged the manuscript to have reached a level acceptable for publication in this journal.
Author Response
Comment 1: The revised manuscript has been appropriately revised according to the reviewers' comments. I judged the manuscript to have reached a level acceptable for publication in this journal.
Response 1: Thank you very much for your appreciation of our manuscript.
Reviewer 2 Report
Comments and Suggestions for Authors
Title: As the authors include the expertise level of the players (amateur level) in the description of the sample (line 124), please add this characteristic in the manuscript title.
Aims: authors should select only one verb to formulate the first objective.
procedures
Please don’t describe the procedure of the inter-rater reliability in the statistical section. Move it before.
Results:
Please improve the resolution of Figure 2
For future review reports, we recommend that authors specify the lines where changes were applied to the manuscript. This practice makes the review process more efficient.
Author Response
Comment 1: Title: As the authors include the expertise level of the players (amateur level) in the description of the sample (line 124), please add this characteristic in the manuscript title.
Response 1: Thank you for your comment. We added this characteristic to the title
Comment 2: Aims: authors should select only one verb to formulate the first objective.
Response 2: Thank you for your comment. We chose only one verb on the first objective (line 94).
Comment 3: Procedures. Please don’t describe the procedure of the inter-rater reliability in the statistical section. Move it before.
Response 3: Thank you for your comment. We believe that the information should stay in the Statistical analysis section, since it’s in this section that interpretation of the test is presented. However, we have included the information regarding the inter-rater reliability in the data collection and analysis section (lines 183-185)
Comment 4: Results: Please improve the resolution of Figure 2
Response 4: Thank you for your comment. We included a new figure, more enlarged and with improved resolution.
Comment 5: For future review reports, we recommend that authors specify the lines where changes were applied to the manuscript. This practice makes the review process more efficient.
Response 5: Thank you for your comment. In previous responses to reviewers comments we did not include line number in the changes made because the information from the editors mentions “Highlight any revisions to the manuscript, so editors and reviewers can see any changes made.” We have added line number in the present response to your comments.